# Acceptability of Prednisolone in an Open-Label Randomised Cross-Over Study—Focus on Formulation in Children

**DOI:** 10.3390/children9081236

**Published:** 2022-08-16

**Authors:** Sissel Sundell Haslund-Krog, Inger Merete Jørgensen, Kim Dalhoff, Helle Holst

**Affiliations:** 1Department of Clinical Pharmacology, Bispebjerg and Frederiksberg University Hospital, 2400 Copenhagen, Denmark; 2Faculty of Health and Medical Sciences, Department of Clinical Medicine, University of Copenhagen, 1165 Copenhagen, Denmark; 3Department of Paediatrics and Adolescents, Nordsjællands University Hospital, 3400 Hillerød, Denmark

**Keywords:** acceptability, paediatric medicine, drug formulations

## Abstract

Developing acceptable medicines for children is a complicated task. Several factors must be considered, including age, physiology, texture preference, formulation, and legal framework among others. In the development of new paediatric medicines, these factors are assessed. However, for older medicines, e.g., prednisolone, acceptability is still a challenge. This study was an open-label randomised three-arm cross-over study investigating different formulations of prednisolone (crushed tablets, whole tablets, and oral solution) in paediatric patients with asthma and asthma-like symptoms. Participants were randomised into two different formulations on two consecutive days. For each formulation, the child or caregiver was asked to evaluate acceptability using a modified five-point Wong Baker Face scale. An analysis of variance (ANOVA) model was used to test for significance. For the 41 children, included mean age was 4.7 years (SD ± 3.6), and mean weight was 21 kg (SD ± 10.8). Sixty-one percent were boys. The participants were divided accordingly into three age groups: 6 to 23 months (N = 11), 2 to 5 years (N = 14), and 6–11 years (N = 16). The overall acceptability was low, with only 23 out of 71 scores rating the treatment either 1 or 2 (32%). The ANOVA test showed a significant difference in acceptability score between crushed tablets and whole tablets (*p* < 0.003). The mean acceptability score for the crushed tablet was the least favourable at 3.9 compared to oral solution (3.1), oro-dispersible tablet (2.8), and whole tablets (2.4). This is problematic in long-term treatment and for the youngest children who cannot swallow tablets. The improvement of age-appropriate and acceptable formulations is necessary.

## 1. Introduction

It is a complicated task to make acceptable medicines for children and a prerequisite to ensure adherence to drug prescriptions and successful treatment outcomes. In paediatric drug development, several factors must be considered and are partly described here. Age and physiology play a significant role when targeting neonates, toddlers, school children, and adolescents. Many medicines are age-specific or administered according to body weight. Body weight is arbitrary as it seldom considers physiological parameters, such as renal or hepatic function, the volume of distribution, lipophilicity, saliva flow rate, gastrointestinal pH or enzyme activity, skin barrier, etc. [1]. The age-related development of these physiological processes can complicate medicine use. Clinical examples are, for example, specific side effects of prednisolone in different age groups or increased dermal absorption and systemic exposure following the topical application of prednisolone in neonates [2,3].

Further, the cognitive abilities of a certain age determine the ability to swallow, taste, and smell. Additionally, the texture preference (dosage form, e.g., solids, powders, suspensions, liquids, and topicals) as well as the choice of formulation (e.g., intravenous, oral, buccal) play significant roles [1,4]. Among 18 parents of HIV-positive children between 4 and 14.5 years of age receiving antiretroviral therapy, specific features contributing to nonadherence were reported in a preliminary survey by the parents. More than 78% had difficulty with the treatment regimen, half of whom attributed this to the taste of the medication [5]. The development of an age-appropriate formulation also requires compatible and stable ingredients and appropriate excipient use. Parabens are used as excipients in, e.g., oral solutions of prednisolone; this can be problematic since propyl-paraben, in particular, may cause adverse reproductive effects in young rats [6]. Parabens have been measured in dried blood samples from neonates who were exposed to several paraben-containing medicines [6]. Exposure limits for excipients in paediatric medicines are currently an area of interest for regulatory stakeholders who dictate the legal framework for paediatric drug development [7].

While many new products designed specifically for children are available, there are still challenges when treating children with old medicinal products, e.g., prednisolone, which has been marketed for more than fifty years. Different children prefer different formulations; e.g., a young child would often favour an oral solution over a tablet, which results in the use of several formulations in paediatric wards. For example, prednisolone is administered as an oral solution, crushed tablets, or whole tablets much to the preference of the child and/or caregiver.

The best methodology to test acceptability in the paediatric population is yet to be determined. In 2017, a review was published stating that hedonic face scales or visual analogue scales (VAS scale) were used most frequently to evaluate acceptability [8]. The criteria for acceptability should always be stated, although consensus regarding the scores is lacking. It has been proposed that the two most positive scores (e.g., 1 or 2) on a 5-point scale are acceptable [8]. A threshold of 80% of acceptance (e.g., acceptability score of 1 or 2) in the population examined is the current standard requirement [8]. A more patient- and caregiver-centred questionnaire (Paediatric Oral Medicines Acceptability Questionnaire) has recently been developed [9]. The questionnaire is yet to be validated but can be used in drug development to assess the acceptability of oral formulations [9]. We can expect further work in this area in search of the best methodology [8,9]. 

So far, very few studies (four studies were found in PubMed and clinical trial.gov) have investigated the acceptability of prednisolone in children with asthma [10,11,12,13]. The number of participants varied from 18 to 255 children from 3 months to 16 years of age. Two studies used a VAS scale [10,12], and two studies a five-point face scale [11,13]. Three studies compared two different formulations [10,12,13] and one study compared five [11]; see Appendix A.

The Pharmacokinetics Of Prednisolone in Children (POP child) study aimed to characterise the bioavailability and pharmacokinetic profile of four oral prednisolone formulations and, as a secondary objective, examine the acceptability of the medicines administered. 

## 2. Materials and Methods

This study was an open-label randomised 3-arm cross-over pharmacokinetic trial investigating different formulations of prednisolone in paediatric patients with asthma and asthma-like symptoms. This study was monitored by the GCP unit at the University of Copenhagen, Denmark.

### 2.1. Participants

Children between 6 months and 11 years of age with asthma and asthma-like symptoms and intended treatment with per oral prednisolone were eligible for inclusion. In total, 41 children were included. 

The participants were stratified during randomisation into 3 age groups i.e., 6–23 months, 2–5 years, and 6–11 years. This was carried out to ensure that the youngest were represented in the study.

### 2.2. Study Design

The primary objective of the POP child study was to examine the bioavailability of different prednisolone formulations, i.e., crushed tablets, oral solution, and oro-dispersible tablets, to characterise their relative bioavailability compared to standard whole tablets (controls) in children with asthma-like symptoms. One of the main secondary objectives was to examine the acceptability of the medicines administered; see the published protocol article for further details [14].

Participants were randomised into two groups using different formulations on two consecutive days (Figure 1). For each formulation, the child or caregiver was asked to evaluate acceptability using a modified five-point Wong Baker Face scale, from positive (1) to negative (5); for details, see Figure 2 and [14]. This ideally results in two acceptability scores per patient. If the child was not capable of using the face scale (typically <6 years), the caregiver was asked to evaluate acceptability. The Wong Baker Face Scale is, as mentioned, most frequently used to evaluate acceptability [8], but none of the scales are validated and no consensus exists on which one to use [1,8].

The POP child study was designed mimicking the clinical use of various prednisolone formulations in treating paediatric patients with asthma-like symptoms following a treatment algorithm of 1 mg/kg (one dose per day) over three days; thus, we could only complete two days per child. In total, 4 formulations were examined, i.e., whole tablets, crushed tablets, oro-dispersible tablets, and oral solution. Whole tablets (controls) could only be administered to children >6 years who could swallow tablets and they received whole tablets both days. Tablets came in 5 mg and 25 mg (with splitting notch) and oro-dispersible tablets in 5 mg and 20 mg (without splitting notch). Supply problems with the oro-dispersible tablets made it necessary to continue this study with only 3 formulations (Figure 1), which deviated from the original protocol.

We also chose to include two strengths of the prednisolone solution, i.e., 5 mg/mL (≤10 kg) and 20 mg/mL (>10 kg), to make sure the volumes used were measurable (volumes below 0.5 mL are more difficult to measure). Both were extemporaneous solutions with no excipients except cherry flavour. Dosing examples are given in Table 1. The complete dosing table is available upon request. As seen in Table 1, the number of tablets and volumes did not always accommodate each weight class. Participants were allowed to mix, e.g., the crushed tablets with yogurt or ice and were offered water to wash down the tablets after administration. In addition to the acceptability, we collected data on age, weight, height, ethnicity, sex, prednisolone time concentrations, co-medication, and adverse events. Only parameters relevant to the acceptability outcome are reported here.

### 2.3. Statistical Analysis

Study data were collected and managed using Research Electronic Data Capture (REDCap) tools hosted in the Capital Region of Denmark [15,16]. For further analysis and visualisation, the data were exported to R 4.0.1, R Foundation for Statistical Computing, Vienna, Austria. Categorical data were presented using percentages and counts. Continuous variables were presented using mean and standard deviation (SD). For the repeated measurement Analysis of Variance (ANOVA), ‘study id’ was set as random effect and acceptability and formulation as fixed effects, and the whole tablet was the reference. Both age and sex were tested for significance in the ANOVA. The ANOVA was chosen since more than two formulations were tested [8] and since it has been used before for the Wong Baker Face Scale [17]. A *p*-value ≤ 0.05 was considered statistically significant.

### 2.4. Sample Size

Due to sparse data in the literature on the acceptability of prednisolone formulations in children, and since acceptability was a secondary outcome, sample size calculations were not feasible. In accordance with the EMA guideline on the investigation of bioequivalence, the number of subjects to be included in a study should preferably be based on sample size calculations and not be less than 12 (3).

## 3. Results

Eighty-three children were screened, and 41 patients were included. Fifteen wanted to participate but could not give consent since one parent was at home or at work, 15 did not wish to participate and 12 were not included for other reasons (e.g., early discharge, inclusion criteria not fulfilled). Thirty-eight completed day 1 and 33 completed day 2 due to, e.g., withdrawal of consent or early discharge.

For the 41 children included, mean age was 4.7 years (SD ± 3.6), mean weight was 21.1 kg (SD ± 10.8). Sixty-one percent were boys (as per Table 2). The participants were divided accordingly in the three age-groups: 6 to 23 months (N = 11), 2 to 5 years (N = 14), 6–11 years (N = 16), Table 2.

The number of patients receiving and rating acceptability for crushed tablets was: 22; for the oral solution: 26; for the oro-dispersible tablets: 9; and for the whole tablets: 14.

### Acceptability

In total, 71 Wong Baker Face scale scores were collected for two consecutive days (38 on Day 1 and 33 on Day 2). The overall acceptability was low with only 23 out 71 scores rating the treatment either 1 or 2 (32%). Regarding the rating of the two individual days, 9/38 (24%) patients rated either a 1 or 2 for Day 1, and 14/33 (42%) rated a 1 or 2 for Day 2.

The ANOVA test showed a significant difference in acceptability score between crushed tablets and whole tablets (*p* < 0.003). The mean acceptability score for the crushed tablet was the least favourable at 3.9 compared to oral solution (3.1), oro-dispersible tablet (2.8), and whole tablets (2.4) (Figure 3). There was no significant difference between the acceptability of whole tablets, oral solution (*p* = 0.13), and oro-dispersible tablets (*p* = 0.47). If we changed the reference group in the ANOVA model to the crushed tablets, then all other formulations were significantly different with regard to acceptability. However, in the protocol [14], the whole tablets were pre-defined as the reference group.

Neither age nor sex was significant in the ANOVA model with *p* = 0.82 and 0.87, respectively.

Sixty-four percent (9 out of 14) of the patients who received whole tablets rated them either 1 or 2. Only 9% (2 out of 22) of the patients who received the crushed tablet rated them either 1 or 2. For the oro-dispersible tablets, the corresponding number was 33% (3 out of 9), and for the oral solution, 35% (9 out of 26) rated it a 1 or 2. Figure 4 shows the acceptability score in relation to the taste categories in the Wong Baker Face scale. Vomiting occurred once by a patient who received the oral solution. Only three patients needed a second attempt of medicine administration.

## 4. Discussion

Our study showed an overall poor acceptability of all four prednisolone formulations. The whole tablets were the most acceptable and crushed tablets the least acceptable. Three different age groups were included; however, neither age nor sex was significant in the ANOVA model. 

In total, 41 patients were enrolled; however, acceptability scores were only obtained in 87% (N = 71). This was mainly due to early discharge (38 rated acceptability on Day 1 and 33 on Day 2) or if the parents withdrew consent. The number of patients rating each formulation was unequal due to supply problems midterm (oro-dispersible tablets), and for the whole tablets due to age exclusivity (<6 years). This is not optimal and is a limitation but reflects the population. In an attempt to minimise the age-formulation effect, our goal was to include a minimum of 12 in each age group, since this is the minimum requirement for a bioequivalence study [18]. We did not succeed in the 6–23 months age group, in which we only included 11 children. Only one other study included children from 3 months of age with no specifications on how many they included [10]. It is well known that young children are more difficult to enrol in clinical studies and generally not well represented [19]. Even though age did not come out as a significant parameter in the ANOVA, the number of patients included might be too low to address this and is a limitation to the study. All relevant age groups should ideally be represented equally in a drug trial. 

Overall, acceptability was low for the four different prednisolone formulations. In the ANOVA, the crushed tablets were the least acceptable and whole tablets the most acceptable. Crushed tablets have previously scored low on acceptability [10], while prednisolone tablets had the lowest palatability score in another study [11]. The oral solution had a median score of 3 in this study, which was comparable to two other studies by Kim et al. (in which the median taste score was 2) and Lucas-Bouwmann et al. (in which the mean VAS score was 3.8 ± 0.5 (SEM)) from 2006 and 2001, respectively. This leaves the youngest age group with a poorly acceptable treatment since they typically cannot swallow tablets. 

Vomiting was observed more frequently (up to 23%) in other studies [10,13] compared to our study and was the primary outcome in one study [13]. The ingestion of prednisolone tablets was associated with more nausea and vomiting compared to syrup and soluble tablets in another study [11]. In the study by Aljebab et al., the treatment regimen was five days and palatability scores improved for all formulations over time [11]. We were not able to demonstrate this due to the shorter treatment period. Notably, the median or mean acceptability scores can be difficult to compare in the studies due to differences in methodology and rating scales, as seen in Appendix A.

In Denmark, we have a relatively small market, which limits the number of available formulations [20], since pharmaceutical companies do not want to pay a marketing fee. This is a huge challenge for the caregivers and for compliance. It also increases the need for extemporaneous preparations, which is a questionable long-term solution. An extemporaneous solution was also used in the study by Lucas-Bouwmann et al. [10], but otherwise marketed formulations seem to be available [11,12,13]. 

The shelf-life of the extemporaneous oral solution was 3 months in our study. This is a limitation for at-home use in chronic diseases. It was also a problem in the clinical study, where we often had to discard half-filled bottles because of limited shelf-life. 

## 5. Conclusions

The overall acceptability of the four prednisolone formulations was poor. This can be problematic when prednisolone is used for long-term treatment in, e.g., chronic diseases or if lower intake due to low acceptability prolongs acute admission. The crushed tablet was the least favourable and the whole tablet the most liked, and the acceptability between the two was significant. This leaves the youngest children without an acceptable formulation, which is surprising when considering that prednisolone has been marketed for more than fifty years. The acceptability between whole tablets, oral solution, and oro-dispersible was not significantly different. Improvements in age-appropriate and acceptable formulations are necessary. Guidelines on how to address acceptability and the proper handling of acceptability data should be more clear and harmonised.

## Figures and Tables

**Figure 1 children-09-01236-f001:**
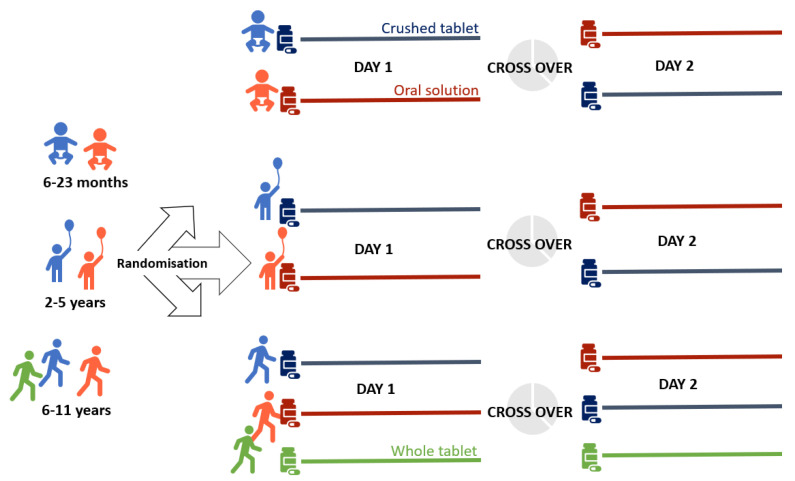
Study design. In the initial design, all participants could be randomised into the oro-dispersible tablet group. When we no longer could procure these tablets, the second design was as pictured in Figure 1. A standard design would have had 4 days for each participant testing all 4 formulations of prednisolone in the initial design and, correspondingly, 3 formulations in the second design.

**Figure 2 children-09-01236-f002:**
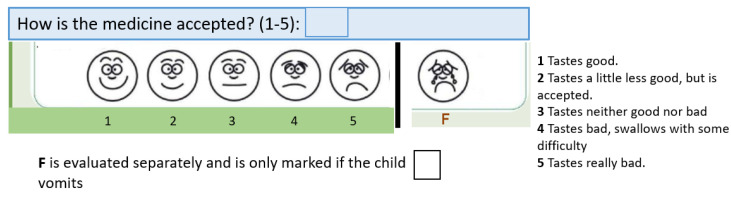
Wong Baker Face scale, as reported in [14].

**Figure 3 children-09-01236-f003:**
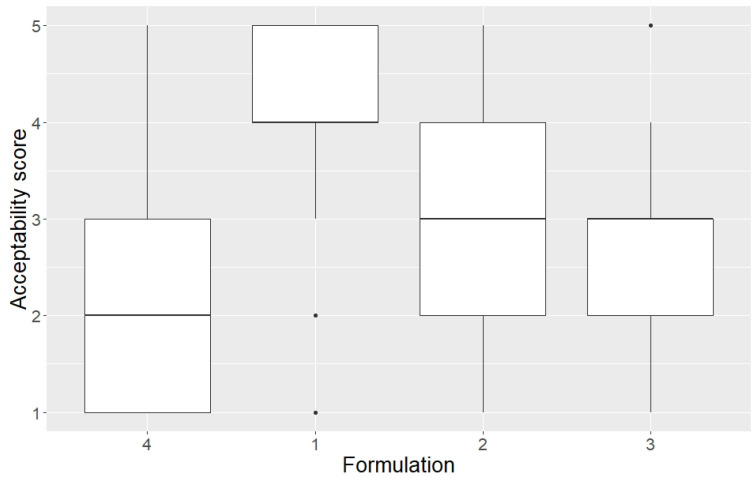
Boxplot with the median acceptability scores for four prednisolone formulations. Minimum, maximum and interquartile range are also shown. Formulations: 1 = Crushed tablets; 2 = Oral solution; 3 = Oro-dispersible tablet; 4 = Whole tablet (reference group).

**Figure 4 children-09-01236-f004:**
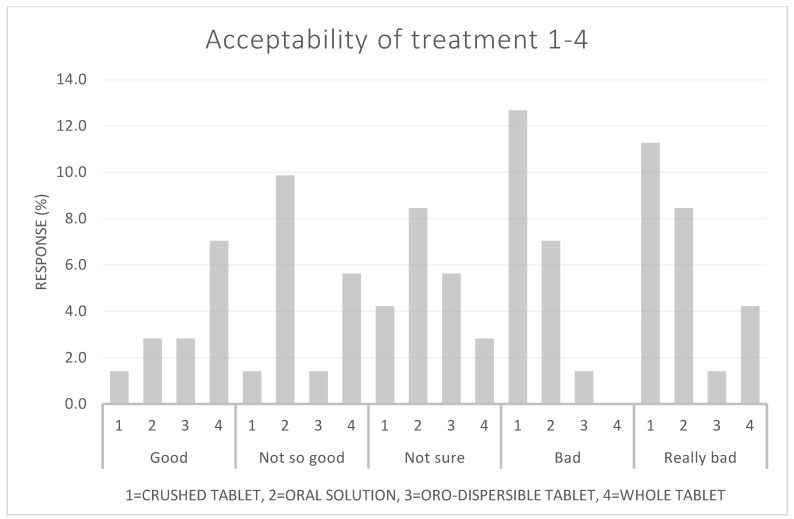
The four prednisolone formulations (1–4) in relation to acceptability scores in percentage. Acceptability scores in each category of the Wong Baker Face scale from ‘Good’ to ‘Really Bad’.

**Table 1 children-09-01236-t001:** Dosing examples.

Weight	Crushed or Whole Tablet Prednisolon	Oro-Dispersible Tablets SOLUPRED^®^	Oral Solution Prednisolon 5 mg/mL	Oral Solution Prednisolon 20 mg/mL
9 kg	5 + 5 mg	5 + 5 mg	1.8 mL	
10 kg	5 + 5 mg	5 + 5 mg	2.0 mL	
11 kg	5 + 5 mg	5 + 5 mg		0.6 mL
15 kg	5 + 5 + 5 mg	5 + 5 + 5 mg		0.8 mL
20 kg	12.5 + 5 + 2.5 mg	20 mg		1 mL

**Table 2 children-09-01236-t002:** Demographic data.

	Number of Patients	N (%)	Mean ± SD	Median (Range)
Age in years	N = 41		4.7 ± 3.6	4.0 (0–11)
Age groups	N = 41	6–23 months: 12 (28.6) 2–5 years: 14 (33.8) 6–11 years: 16 (38.1)		
Gender	N = 41	Male: 25 (61) Female: 16 (39)		
Race	N = 41	Caucasian: 31 (77.5) Non-Caucasian: 9 (22.5)		
Height in m at inclusion	N = 34		1.16 ± 0.23	1.16 (0.78–1.56)
Weight in kg at inclusion	N = 41		21.1 ± 10.8	17.6 (8.2–44.5)

## Data Availability

Data can be made available upon reasonable request.

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
