# Peer review of "Acceptability of Prednisolone in an Open-Label Randomised Cross-Over Study—Focus on Formulation in Children"

_children, 2022, doi:10.3390/children9081236_

Round 1

Reviewer 1 Report

This study assesses the acceptability of different prednisolone formulations in paediatric patients. I would suggest that the following amendment be made:

1.       Present the abstract as one continuous paragraph (i.e., remove subheadings like background, conclusion etc.)

2.       Line 43 – 44: the sentence “several factors must be … is hanging and should be deleted to rephrased.

3.       Line 49: …complicate medicine and not medicinal

4.       Line 50 – 51: reword the sentence and place references at the end of the statement

5.       Line 61 – 62: Sentence/example provided is out of place. Best to edit it and make it complement other sentences in the paragraph

6.       Line 67-68: “The child’s formulation preferences…. Is not clear and sentence needs to be improved upon

7.       Line 85: What is the VAS scale. It would be best to write the full meaning first before abbreviating

8.       Line 94: Delete one “in”

9.        Table 1 (particularly the headings) need some improvement. Amend Table rows and columns

10.   Line 143 -144: This statement should be part of the main text (e.g., section 2.2. or where you deem fit) not under the table. Please move accordingly

11.   Line 151: ANOVA needs to be written in full first before abbreviation

12.   Please make Table 2 better

13.   Section 3.1; Line 177 – If 41 patients were included in the study, how did you arrive at 71 Wong Baker Face Scale scores….? Please make this clearer in the method section as well

14.   Lines 181-189: The reference/standard value that makes p-value significant or not needs to be mentioned somewhere in the method (before discussion)

15.   Lines 200-201: seem to be out of place. Please double check and move to a more appropriate spot in the manuscript

16.   Line 202 – 206, Figure legend should be reworded in past tense

17.   Figure 3 – Line 209 – 211: Improve on all graph labels. I would suggest that the words “acceptability” and “formulation” should be boldened, upper case first letter and increase font size for legibility. Increase the font size of the numbers

18.   Line 237-238 needs to be improved. It is not clear

19.   Line 237: remove extra space between “2006” and “and”…

20.   I would recommend that the conclusion is presented as one paragraph

Author Response

Thanks for your comments. Please see the changes in the revision.

Reviewer 2 Report

The authors need to cite the protocol for this study which has been published. The Methods can then be shortened dramatically, only stating a summary and deviations.

The authors also need to read the instructions to authors and delete sections of the paper re formatting (see paragraph 0)

Round 2

Reviewer 1 Report

None

Author Response

Thank you for revising the manuscript. There were no comments to address, however, the English language in the manuscript has been carefully edited (see track changes) and unnecessary words/lines have been deleted.

Reviewer 2 Report

The paper has improved. However, the results need to be presented more clearly. Fig 3 has an error in the PDF. The authors need to point out the low numbers of patients in their discussion as a limitation of their findings
